# Treatment with Angiotensin-(1-7) Prevents Development of Oral Papilloma Induced in K-ras Transgenic Mice

**DOI:** 10.3390/ijms23073642

**Published:** 2022-03-26

**Authors:** Carolina Schere-Levy, Melisa Suberbordes, Darío M. Ferri, Marina Ayre, Albana Gattelli, Edith C. Kordon, Ana R. Raimondi, Thomas Walther

**Affiliations:** 1Facultad de Ciencias Exactas y Naturales, Universidad de Buenos Aires (UBA), Buenos Aires C1428EGA, Argentina; melissagunner@hotmail.com (M.S.); dario_ferri@hotmail.com (D.M.F.); maru.ayre@gmail.com (M.A.); albanagattelli@gmail.com (A.G.); ekordon@gmail.com (E.C.K.); rosar4071@gmail.com (A.R.R.); 2IFIBYNE-CONICET-UBA, Instituto de Fisiología, Biología Molecular y Neurociencias (IFIBYNE), Ciudad Universitaria, CABA, Buenos Aires C1428EGA, Argentina; 3Department of Pharmacology and Therapeutics, School of Medicine and School of Pharmacy, University College Cork, T12 YN60 Cork, Ireland; walthert@uni-greifswald.de; 4Institute of Medical Biochemistry and Molecular Biology, University Medicine Greifswald, 17475 Greifswald, Germany; 5Xitra Therapeutics GmbH, Berlin-Buch, 13125 Berlin, Germany

**Keywords:** Ang-(1-7), K-ras, mTor, oral cancer, papilloma, tumor

## Abstract

Oral Squamous Cell Carcinoma (OSCC) is the most common malignant cancer affecting the oral cavity. It is characterized by high morbidity and very few therapeutic options. Angiotensin (Ang)-(1-7) is a biologically active heptapeptide, generated predominantly from AngII (Ang-(1-8)) by the enzymatic activity of angiotensin-converting enzyme 2 (ACE 2). Previous studies have shown that Ang-(1-7) counterbalances AngII pro-tumorigenic actions in different pathophysiological settings, exhibiting antiproliferative and anti-angiogenic properties in cancer cells. However, the prevailing effects of Ang-(1-7) in the oral epithelium have not been established in vivo. Here, we used an inducible oral-specific mouse model, where the expression of a tamoxifen-inducible Cre recombinase (CreER^tam^), which is under the control of the cytokeratin 14 promoter (K14-CreER^tam^), induces the expression of the K-ras oncogenic variant KrasG12D (LSLK-ras^G12D^). These mice develop highly proliferative squamous papilloma in the oral cavity and hyperplasia exclusively in oral mucosa within one month after tamoxifen treatment. Ang-(1-7) treated mice showed a reduced papilloma development accompanied by a significant reduction in cell proliferation and a decrease in pS6 positivity, the most downstream target of the PI3K/Akt/mTOR signaling route in oral papilloma. These results suggest that Ang-(1-7) may be a novel therapeutic target for OSCC.

## 1. Introduction

The renin-angiotensin system (RAS) was originally described as an endocrine system, in which the substrate protein angiotensinogen is sequentially cleaved by two peptidases, renin and angiotensin-converting enzyme (ACE), to form the biologically active octapeptide angiotensin II (AngII) [1]. AngII, the main effector peptide of the RAS, has been implicated in multiple aspects of tumor progression such as proliferation, migration, invasion, angiogenesis, and metastasis [2,3,4]. A large body of evidence supports the existence of a tissue-specific RAS that fulfills particular functions in each tissue, where AngII is produced locally, exerting various pathophysiological effects [5]. Ang-(1-7) is a biologically active heptapeptide, generated primarily through conversion of AngII by the angiotensin-converting enzyme 2 (ACE2), a peptidase, which also acts as the receptor for SARS-CoV-2 to allow the virus to enter human lung epithelial cells [6]. Ang-(1-7) attracted special attention over the last two decades for showing the critical ability to counteract many actions of AngII in different pathological settings [7,8,9]. Furthermore, we previously identified the two receptors which mediate the effects of Ang-(1-7): the G protein-coupled receptor Mas [7] and the G protein-coupled receptor MrgD [10]. Thus, while the classical arm of the RAS with ACE/AngII/AT1 causes detrimental features in several tissues, the system also contains a beneficial arm with ACE2/Ang-(1-7)/Mas and MrgD (Figure 1).

Previous studies from our group have shown that Ang-(1-7) counteracts tumorigenic effects triggered by AngII in triple-negative breast cancer cells [9]. In vivo studies have indicated that the administration of Ang-(1-7) reduces lung, prostate, nasopharyngeal and hepatocellular cancer [11,12]. Our group recently found that VEGF receptor tyrosine kinase inhibitors (TKI), such as Sunitinib or Axitinib, significantly downregulate ACE2 expression in tumor cells, and that the reduction in ACE2 activity is the driver for the development of resistance to such VEGF signaling inhibitors in various preclinical models of kidney cancer [13]. From the translational perspective, the key finding was that the resistance to Sunitinib or Axitinib could be reversed by treating mice with Ang-(1-7), thus compensating for ACE2 deficiency, resulting in less tumor growth and improved overall survival [13]. Thus, the treatment with Ang-(1-7), directly or as co-treatment with a VEGF receptor TKI, might be an attractive therapeutic strategy to reverse tumor progression and angiogenesis [14].

Oral squamous cell carcinoma (OSCC) is the eighth most common cancer worldwide [15]. Cancers originating from the mucosal epithelial lining of the oral cavity, larynx, and pharynx, are collectively called head and neck squamous cell carcinoma (HNSCC) [15]. In total, 300,000 new cases of HNSCC are diagnosed worldwide each year, and the prevalence has significantly increased in recent years, particularly amongst the younger population [16]. Risk factors for HNSCC are usually lifestyle-associated and include smoking, excessive alcohol consumption, and betel nut chewing [16]. This disease is associated with significant mortality and morbidity rates, despite the large amount of research and advances made in the field of oncology and surgery. Only 40–50% of patients with HNSCC will survive >5 years. Therefore, an alternative treatment is urgently needed [15].

The aberrant activation of the PI3K/mTOR signaling circuitry is one of the most frequently dysregulated signaling events in HNSCC [17]. In this sense, it has been reported that Ang-(1-7), which also has anti-inflammatory properties, plays an important role in the inactivation of the PI3K/Akt/mTOR pathway in several tissues [12,18]. Reports show that Ang-(1-7) has a growth-reducing role in nasopharyngeal tumors [12,19]. It has been shown that AT1 expression was associated with OSCC progression [20] and that Ang-(1-7) counteracts the tumor effects triggered by AngII in HNSCC cells [21]. It has been widely described that the AngII/ACE/AT1 pathway induces tumor progression and metastasis in various tissues [2,3,4]. However, treatment with ACE inhibitors (ACEi) or AT1 antagonists (ARBs) does not seem to be very effective in inhibiting tumor growth. Therefore, the beneficial arm of the RAS could be more efficient than inhibiting/blocking the detrimental arm of the system. This was also supported by studies in other tumors treated with Ang-(1-7), including our work in renal cell carcinoma [13]. However, the in vivo effects of Ang-(1-7) in the oral cavity have not been explored thus far.

We previously showed that mice expressing a tamoxifen-inducible Cre recombinase under the control of the cytokeratin 14 (K14) promoter, finally expressing an oncogenic mutation of K-ras (LSL-K-ras^G12D^ mice), develop squamous papilloma in the oral cavity within 1 month from tamoxifen treatment [22].

In this study, we have analyzed the effect of Ang-(1-7) treatment on the development of K-ras-driven oral papilloma.

## 2. Results

### 2.1. Expression of Components of The Renin-Angiotensin System in Head and Neck Squamous Cell Carcinoma Originated in Distinct Head and Neck Sites

Detrimental effects of the classical arm of the RAS, including ACE, AngII, and the AngII receptor type 1 (AT1), in cancer progression, angiogenesis and metastasis have been extensively documented, and their expression in the oral cavity has been demonstrated [12,23]. In our first approach, we analyzed these RAS components in the HNSCC TCGA database. mRNA of angiotensinogen, the precursor protein of AngII, ACE, and the AT1 receptor could be detected in HNSCC regardless of where the tumor originated. There were no significant differences in the expression levels when tumors developed in different head and neck cavities (Appendix A).

Much less is known about the expression of the beneficial arm of the RAS in HNSCC. Therefore, we searched the TCGA database for expression levels of ACE2 and the Ang-(1-7) receptors Mas and MrgD. Similar to ACE, mRNA of the peptidase ACE2 could be detected in all tumors, with no differences in the expression intensity depending on the tumor’s location (Figure 2A). The expression of Mas was very low in a few tumors originating from buccal mucosa, alveolar ridge, hard palate, and lip, and below detection levels in the majority of all tumors (Figure 2B). MrgD (Figure 2C) expression was also often not detectable, but, where measurable, regions were distinct from Mas, as it was only detectable in the larynx and floor of the mouth.

### 2.2. Angiotensin-(1-7) Prevents Growth of Oral Lesions Induced by K-ras Activation in K14-CreER^tam^/LSL-K-ras^G12D/+^ Mice

Next, we aimed to evaluate the potential therapeutic role of Ang-(1-7) in oral cavity tumors in a transgenic preclinical model of the disease. The animal model is described in detail in the Materials and Methods section, and the treatment regimen in our in vivo experiment is shown in Figure 3A. As previously described, our K14-CreER^tam^/LSL-K-ras^G12D/+^ mice developed severe squamous cell papilloma in the oral mucosa within 1 month after tamoxifen treatment [22]. As shown in Figure 3B, tamoxifen induction led to the development of large exophytic squamous cell papilloma, which were already macroscopically very prominent. Squamous papilloma are exophytic tumors of the oral mucosa. These papilloma were characterized at the microscopic level by hyperplastic epithelium covering thin stromal finger-like projections that grow out of the oral squamous epithelium (Figure 3C). In contrast, the subcutaneous injection of Ang-(1-7) (0.5 mg/kg/d) into such tamoxifen-treated transgenic mice led to much smaller papilloma as seen macroscopically (Figure 3D,E), whereby the smaller papilloma still carried typical histological features of a papilloma (Figure 3F,G).

### 2.3. Angiotensin-(1-7) Prevents Cell Proliferation and Activation of the mTOR Pathway

In agreement with previous results, the transgenic control mice showed high levels of cell proliferation in oral papilloma [22,24], as illustrated by positive immunostaining of cell mitosis with a p-histone H3 (Ser 10) antibody (Figure 4A). However, animals treated with Ang-(1-7) showed significantly less proliferation in oral papilloma (Figure 4B,C).

The aberrant activation of the PI3K/mTOR signaling circuitry is one of the most frequently dysregulated signaling events in HNSCC [17]. It has been reported that Ang-(1-7), a peptide with anti-inflammatory properties, plays an important role in the inactivation of the PI3K/Akt/mTOR pathway in several tissues [12,18]. Therefore, we tested whether Ang-(1-7) affects the mTOR signaling pathway. pS6, the most downstream target of the PI3K/Akt/mTOR signaling route, was detected by immunostaining. As shown in Figure 4, we found a decrease in the level of pS6 in oral papilloma of Ang-(1-7)-treated mice (Figure 4E,G) in contrast to the more pronounced signals in vehicle-treated animals (Figure 4D,F). These results suggest that mTOR inhibition could be a potential pathway involved in this protection process by Ang-(1-7).

## 3. Discussion

AngII, the main effector peptide of the renin-angiotensin system, is a pro-inflammatory octapeptide that has been implicated in multiple aspects of cancer progression such as proliferation, migration, invasion, angiogenesis, and metastasis, with an increasing number of studies showing a relationship between ACE/AT1/AngII dysregulation and carcinogenesis [4,25]. While initial studies showed expressions of members of the detrimental axis, such as ACE and AT1 in primary head and neck cutaneous SCC and oral cavity SCC of different subsites [23], we demonstrate that the expression is relatively high but is not regulated or dependent on the origin of the OSCC.

Two decades ago, using a sensitive method of mRNA detection/RNAse protection assay, we were the first to investigate Mas expression in the tongue tissue of healthy mice [26] and found traces of mRNA, but much weaker signals than in tissue, with the highest expression, testis, and forebrain. Using current mRNA profiles from the HNSCC TCGA database, we show that also under pathophysiological conditions, expression of Mas in HNSCC tissue, regardless of the origin, is very weak or cannot be detected at all. This also counts for the second heptapeptide receptor, MrgD. There is little correlation between the detectable levels of MrgD mRNA and those of Mas.

Ang-(1-7) counteracts undesirable actions of AngII in different pathophysiological settings [9,27]. It has been shown that Ang-(1-7) plays an important role in preventing cancer progression in a variety of tumors, including lung carcinoma [11,18], prostate cancer [28], breast cancer [29,30], and hepatocellular carcinoma [31]. Few studies have shown a regulatory role of Ang-(1-7) in nasopharyngeal carcinoma (NPC), where expression of Mas was also described [12,19]. The authors also demonstrated that Ang-(1-7) might reduce tumor growth in NPC xenografts by downregulating the PI3K/Akt/mTOR signaling pathway. Another group assessed the association between the use of AT1 receptor blockers (ARBs) and the survival rates in patients with oral SCC of different stages, showing increased overall survival (OS) rate of ARB users after 180 days of treatment [32]. Although this suggested an important therapeutic role of Ang-(1-7), since ARBs increase the concentration of the heptapeptide, the effects of Ang-(1-7) in the oral cavity have not been explored. Our results clearly show that the severe and visible premalignant lesions in the oral mucosa of untreated mice were significantly smaller in volume when treated with Ang-(1-7). This was accompanied by a significant reduction in proliferating cells in those papilloma.

Activating the AKT/mTOR signaling pathway resulting in pS6 accumulation seems to represent early events in HNSCC carcinogenesis [22,33]. It has been demonstrated that Ang-(1-7) inhibits carcinogenesis via mTOR signaling in nasopharyngeal carcinoma and lung injury [12,18]. A few years ago, we reported that Ang-(1-7) blocks AKT activation and VEGF expression induced by AngII in triple-negative breast cancer (TNBC) cells [9]. Previously, we have published that treating K14-CreER^tam^/LSL-K-ras^G12D/+^ mice with rapamycin, an mTOR inhibitor, prevented the development of oral papilloma and tongue carcinomas [22]. In this study, we found a diminished expression of pS6, the most downstream target of the PI3K/Akt/mTOR signaling pathway, in lesions from Ang-(1-7) treated mice of the preclinical model of OSCC, in comparison to untreated mice.

Mechanistically, the way the heptapeptide can initiate its beneficial effects might be unclear from our expression profile of Mas and MrgD in human HNSCC tissue, since we found small amounts of Mas and/or MrgD expression in areas of human tumor tissue corresponding to the areas of papilloma formation in our preclinical mouse model. This will be investigated in the future by crossing the transgenic K-ras transgenic mice with Mas and MrgD deficient mice. Additional experiments with the classical Ang-(1-7) antagonist D-Pro^7^-Ang-(1-7) (D-Pro) or D-Ala^7^-Ang-(1-7) (A779) together with Ang-(1-7) will determine whether the beneficial effects of Ang-(1-7) are mediated by its classical receptors or by receptors so far not associated with Ang-(1-7) signaling.

In summary, our results support the potential clinical benefit of targeting mTOR signaling with Ang-(1-7) to treat head and neck squamous cell carcinomas in the oral cavity. As the heptapeptide Ang-(1-7) has already been approved by the Food and Drug Administration for use in various clinical trials [34,35], its use in a cancer-related therapeutic regimen could be quite easily tested.

## 4. Materials and Methods

### 4.1. UCSC Xena Cancer Genomics Browser Analysis

A University of California Santa Cruz (UCSC) Xena browser (https://xenabrowser.net accessed on 12 December 2021) was used to compare the mRNA expression of the various members of the renin-angiotensin system at distinct anatomic locations of Head and Neck Cancer (HNSC). The IlluminaHiSeq dataset TCGA Head and Neck Cancer dataset study (21 datasets, *n* = 604 samples, 566 RNA sequenced) were used for the analyses. Values show the gene-level transcription estimates, as log2 (norm_count +1) transformed RSEM normalized count. Genes are mapped onto the human genome coordinates using UCSC Xena HUGO probeMap.

### 4.2. Transgenic Mice

The K14-CreER^tam^ and LSL-K-ras^G12D^ mouse strains have been described before [22]. Their genetic background was FVB/N–K14-CreER^tam^ mice were crossed with LSL-K-ras^G12D^/+ mice to generate K14-CreER/K-ras^G12D^/+ mice. K14-CreER^tam^ mice were used as hemizygotes in the double line established. Genotyping was performed on tail biopsies by PCR using specific primers [22]. Tamoxifen (Sigma-Aldrich, St. Louis, MO, USA) was administered to 1-month-old animals, 1 mg per mouse per day orally, for five consecutive days. Treatments were started at the time of first tamoxifen administration and continued for 30 days. All mice were examined daily. Control mice received saline solution as a vehicle (*N* = 4) *by* subcutaneous injection every day. Ang-(1-7), manufactured by Biosyntan, Berlin-Buch, Germany, was dissolved in PBS and administered to mice (*N* = 6) through subcutaneous injection at 0.5 mg/kg daily. All experimental procedures were in accordance with institutional (Institutional Animal Care and Use Committee of the Facultad de Ciencias Exactas y Naturales, University of Buenos Aires) and government regulations. All efforts were made to minimize the number of animals used and their suffering by using the commonly accepted 3Rs: Replacement of animals by alternatives wherever possible, Reduction in the number of animals used, and Refinement of experimental conditions and procedures to minimize harm to animals.

### 4.3. Immunohistochemistry (IHC) Staining

For histological, immunohistochemical analysis, tissues were immersed overnight in 10% buffered formalin and then embedded in paraffin. Morphological features were evaluated in hematoxylin-eosin (H&E)-stained slides. Immunohistochemical assays were performed as described previously [22,36]. In short, the samples were incubated overnight at 4 °C with the following antibodies: anti-rabbit phospho-Histone H3 Ser 10 (p-H3) pS10H3 (Cell Signaling Technology, Inc., Beverly, MA, USA) or anti-rabbit phopho-S6 ribosomal protein Ser240/244 (pS6) (Cell Signaling Technology, Inc., Beverly, MA, USA). The specificity of the immunostaining was tested by omission of the primary antiserum. Controls gave negligible background staining (data not shown). For each staining, the quantification was done in each papilloma section, counting positive and negative cells expressed as a percentage (%) of positive cells/field ± SEM. We determined the number of p-H3-positive cells in fields 8–10 via higher magnification (×40) images of the papilloma related to the total number of cells for each field.

### 4.4. Statistical Analysis

We used two-sided two-sample T-tests to compare measurements between treatment and control groups. The results were considered statistically significant if the *p*-value of a statistical hypothesis test was smaller than the significance level of 0.05. Analysis and graphs were calculated by GraphPad Prism 9 (San Diego, CA, USA).

## Figures and Tables

**Figure 1 ijms-23-03642-f001:**
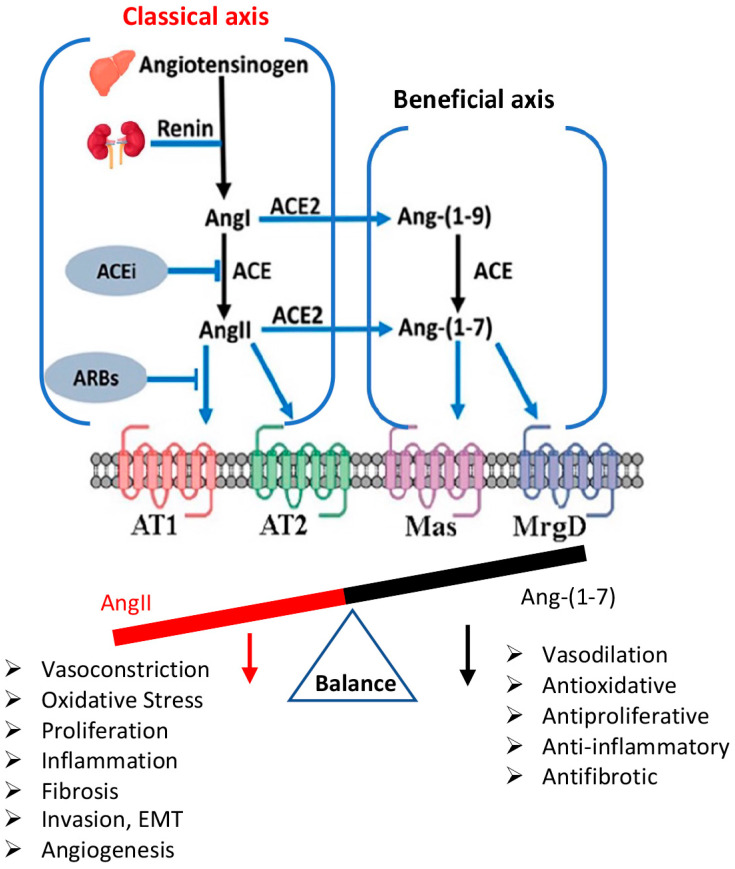
The classical and beneficial arms of RAS. Angiotensinogen is produced by the liver, then hydrolyzed by renin to form angiotensin I (AngI). Next, Ang I is hydrolyzed by angiotensin-converting enzyme (ACE) to produce the octapeptide angiotensin II (AngII). AngII can stimulate two receptors, AngII type 1 receptor (AT1) and AngII type 2 receptor (AT2). ACE2 catalyzes Ang II to generate the heptapeptide angiotensin-(1-7) (Ang-(1-7)), which can also be a product of ACE activity cleaving angiotensin-(1-9) (Ang-(1-9)). Ang (1-7) counteracts the effects of AngII by interacting with the receptors Mas (Mas) and/or MrgD.

**Figure 2 ijms-23-03642-f002:**
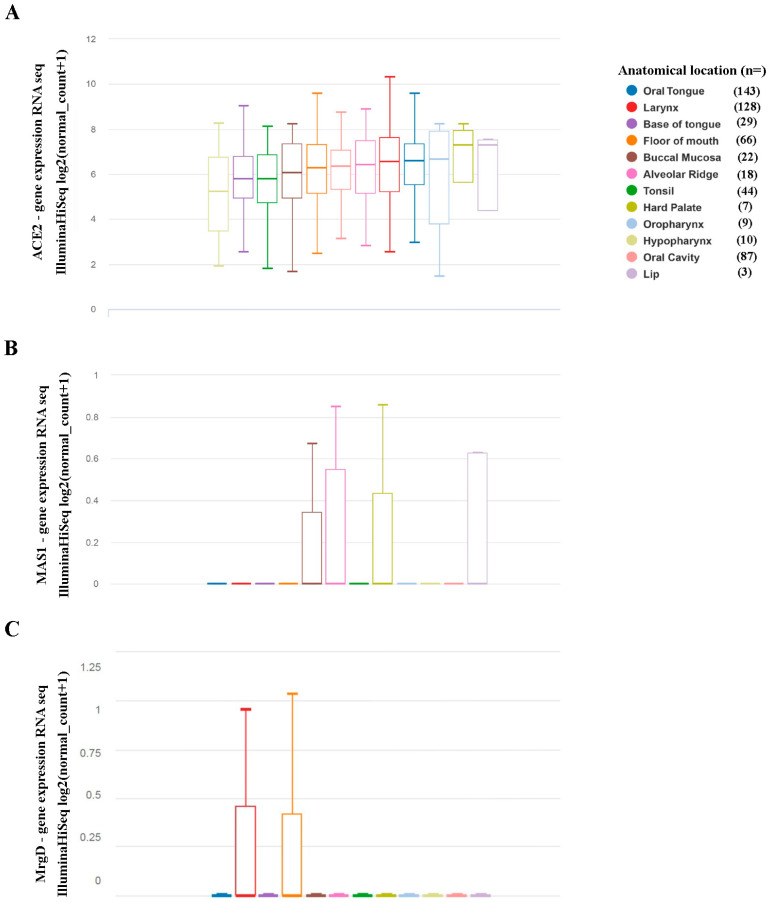
Levels of gene expression of members of the beneficial arm of the renin-angiotensin system in head and neck squamous cell carcinoma (HNSCC). Data from the TCGA HNSCC study (21 datasets, *n* = 604 samples, hence 566 with mRNA analyses) were used to investigate the expression of ACE2 (**A**), Mas (**B**), and MrgD (**C**), along with distinct anatomic locations of HNSCC. Analyses were performed using the UCSC Xena browser.

**Figure 3 ijms-23-03642-f003:**
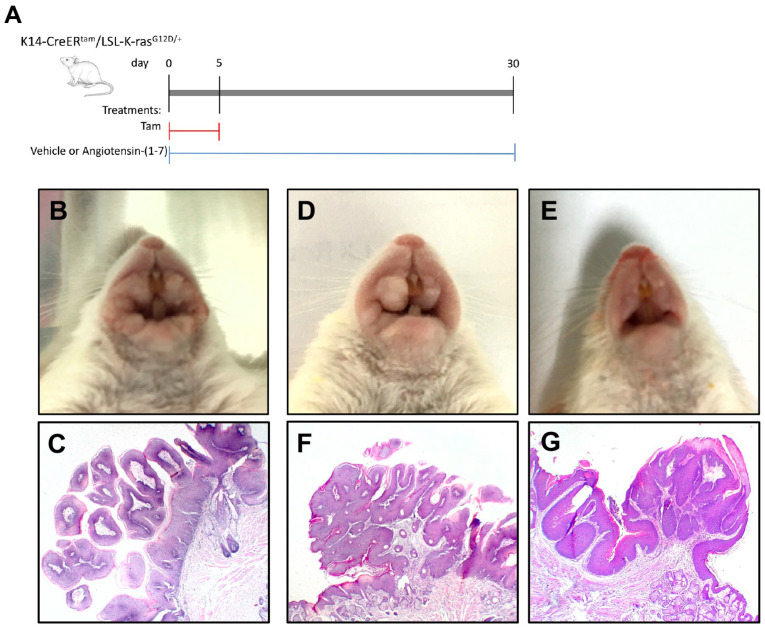
(**A**) Scheme of experimental groups and timeline used in the study. A representative example of K14-CreER^tam^/LSL-K-ras ^G12D/+^ mice treated with vehicle (**B**) after Tam induction as indicated in the upper scheme. (**C**) Histology of an oral squamous cell papilloma from a K14-CreER^tam^/LSL-K-ras ^G12D/+^ mouse upon Tam induction and vehicle treatment for 30 days (*n* = 4 animals). The hyperplastic epithelium covering thin stromal finger-like projections that grow out of the oral squamous epithelium. Two representative examples of K14-CreERtam/LSL-K-ras G12D/+ mice treated with angiotensin-(1-7) (**D**,**E**) and the histology (**F**,**G**) of small oral squamous papilloma from those representative K14-CreERtam/LSL-K-rasG12D/+ mice upon Tam induction and angiotensin-(1-7) treatment during 30 days (*n* = 6 animals). Original magnification: ×4.

**Figure 4 ijms-23-03642-f004:**
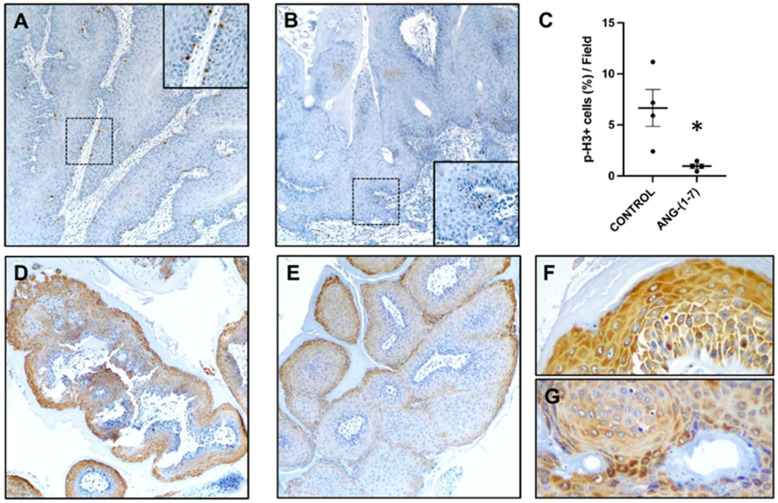
(**A**–**C**): Proliferative status of the oral papilloma from the K14-CreER^tam^/LSL-K-ras ^G12D/+^ mice treated with vehicle or angiotensin-(1-7). (**A**) p-H3 expression in a control mouse. The immunostaining is widely extended in the basal layers (inset). (**B**) p-H3 expression is reduced after angiotensin-(1-7) treatment (detail depicted in the lower inset). (**C**) p-H3 positive cells were counted and quantified as a percentage of total cells. Angiotensin-(1-7) treated tumors showed a significant reduction of p-H3 labeled cells. Student t-test (*n* = 4) * *p* < 0.05. (**D**–**G**) pS6 immunostaining in representatives papilloma developed by K14-CreER^tam^/LSL-K-ras ^G12D/+^ mice treated with vehicle (**D**,**F**) or angiotensin-(1-7) (**E**,**G**). Level of pS6 in the tumor of a mouse treated with the vehicle was more prominent and showed stronger signal intensity than the angiotensin-(1-7) treatment (**E**,**G**). Original magnifications: A,B ×10; D,E ×20; and F,G ×40.

## Data Availability

No further data sets other than those shown have been generated.

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
