# Peer review of "Treatment with Angiotensin-(1-7) Prevents Development of Oral Papilloma Induced in K-ras Transgenic Mice"

_ijms, 2022, doi:10.3390/ijms23073642_

Round 1
Reviewer 1 Report
General comments
In this study, the authors showed that Ang-(1-7) administration reduced oral papilloma through pS6 level suppression in mice with K-ras-induced papillomas. Based on these results, the authors argue that Ang-(1-7) has clinical utility for oral cancer. Regardless of the molecular mechanism, there is no doubt that administration of Ang-(1-7) decreases the tumor size by reducing the number of proliferating cell. Therefore, it may be a candidate as a therapeutic drug for oral cancer. However, in that regard, the report in this paper it is not always novel. The novelty of this paper may have been shown in vivo using transgenic mice that can induce papilloma.
One of the weaknesses of this paper is that the localization of MasR, which is important in ACE2/Ang-(1-7)/MasR axis, is not shown. In addition, although it is logic that lowering pS6 levels downstream of the AKT/mTOR pathway by Ang-(1-7) leads to growth suppression, actual data not necessary strongly support it.
Specific comments
- Mice used in in vivo experiments developed benign papillomas under Ang-(1-7) administration. These data suggest that administration of Ang-(1-7) may not have a suppressive effect on the onset of papilloma itself. If administration of Ang-(1-7) causes a decrease in the number of proliferating cells, it may be better to investigate anti-cancer effects for transplanted malignant cancer tumors.
- Comparing Figure 3EF and Figure 3EG, Ang-(1-7) administration does appear to reduce the pS6 immunostaining level in cancer tissues. However, the difference in pS6 level does not appear to be significant for the basal layer where mitotic cells are located. Including that point, the authors need to add data on the association between pS6 level decline and growth suppression.
- Ang-(1-7) acts on cells via MasR. According to Figure 1B, Mas expression is observed only in some tissues of HNSCC. The authors should be able to deepen their understanding of how Ang-(1-7) works by showing the results of Mas immunostaining results.
Reviewer 2 Report
- The manuscript with the title "Treatment with angiotensin–(1-7) prevents development of oral papilloma induced in K-ras transgenic mice" is well written and deals with an interesting topic. However, I feel that all readers that are not particularly familiar with the details of the RAS system would highly benefit from a figure with an overview showing which enzyme cleaves which molecule in order to have all relevant information at one sight. This would also be very helpful in order to understand the findings presented in Figure 1.
- What is meant by "high power fields of papilloma" in 4.3.?
- What exactly do you mean by the "beneficial arm of RAS"? I guess I know what you want to say however it is not clearly stated in the manuscript.
- I wonder why RAS is so interesting in OSCC? Is anything already known about it? I would include this information in the introduction.
- I think that the information given in lines 127 ff. should also be included in the introduction.
- Generally, some information given in the discussion should be transferred to the introduction in order to introduce the reader into the field.
